# Twelve-month specific IgG response to SARS-CoV-2 receptor-binding domain among COVID-19 convalescent plasma donors in Wuhan

Cesheng Li[1,7], Ding Yu[2,3,7], Xiao Wu[1], Hong Liang[2,3], Zhijun Zhou[1], Yong Xie[1,3], Taojing Li [3], Junzheng Wu [2], Fengping Lu[1], Lu Feng[1], Min Mao[1], Lianzhen Lin[1], Huanhuan Guo[1], Shenglan Yue[1], Feifei Wang[1], Yan Peng[1], Yong Hu[1], Zejun Wang[4], Jianhong Yu[1], Yong Zhang[3], Jia Lu[4], Haoran Ning[1], Huichuan Yang[5], Daoxing Fu[3], Yanlin He[1,3], Dongbo Zhou[3], Tao Du[3], Kai Duan[4], Demei Dong[3], Kun Deng[1], Xia Zou[1], Ya Zhang[1], Rong Zhou[3], Yang Gao[3], Xinxin Zhang[6,8✉] & Xiaoming Yang [5,8✉]

To investigate the duration of humoral immune response in convalescent coronavirus disease 2019 (COVID-19) patients, we conduct a 12-month longitudinal study through collecting a total of 1,782 plasma samples from 869 convalescent plasma donors in Wuhan, China and test specific antibody responses. The results show that positive rate of IgG antibody against receptor-binding domain of spike protein (RBD-IgG) to severe acute respiratory syndrome coronavirus 2 (SARS-CoV-2) in the COVID-19 convalescent plasma donors exceeded 70% for 12 months post diagnosis. The level of RBD-IgG decreases with time, with the titer stabilizing at 35.7% of the initial level by the 9th month. Moreover, male plasma donors produce more RBD-IgG than female, and age of the patients positively correlates with the RBD-IgG titer. A strong positive correlation between RBD-IgG and neutralizing antibody titers is also identified. These results facilitate our understanding of SARS-CoV-2-induced immune memory to promote vaccine and therapy development.

[1] Sinopharm Wuhan Plasma-derived Biotherapies Co., Ltd, Wuhan, China. [2] Chengdu Rongsheng Pharmaceuticals Co., Ltd, Chengdu, China. [3] Beijing Tiantan Biological Products Co., Ltd, Beijing, China. [4] Wuhan Institute of Biological Products Co. Ltd, Wuhan, China. [5] China National Biotec Group Company Limited, Beijing, China. [6] Research Laboratory of Clinical Virology, Ruijin Hospital and Ruijin Hospital North, National Research Center for Translational Medicine, Shanghai Jiao Tong University of Medicine, Shanghai, China. [7] These authors contributed equally: Cesheng Li, Ding Yu. [8] These authors jointly supervised this work: Xinxin Zhang, Xiaoming Yang. ✉email: zhangx@shsmu.edu.cn; yangxiaoming@sinopharm.com

Since the emergence of coronavirus disease 2019 (COVID-19), caused by severe acute respiratory syndrome coronavirus 2 (SARS-CoV-2) in December 2019, the virus has spread rapidly and globally, leading to a pandemic outbreak. As of February 19, 2021, SARS-CoV-2 has infected more than 109 million people worldwide, with the death toll exceeding 2.4 million, and approximately 364,000 newly diagnosed cases are still daily reported (https://covid19.who.int/, accessed February 19, 2021).

Several SARS-CoV-2 vaccines have been approved worldwide, but their longevity of immune protection is still uncertain. Evaluating the durability of the immune response, especially humoral immune response, induced by SARS-CoV-2 is essential to understand the pathogenesis of SARS-CoV-2 and predict the longevity of its vaccine protection, which further facilitates the urgent development of vaccine or therapeutics[1]. In the patients infected with severe acute respiratory syndrome coronavirus 1 (SARS-CoV-1), the specific antibodies against SARS-CoV-1 can last for an average of 2 years, with the positive rate and titer of SARS-CoV-1-specific neutralizing antibodies significantly reduced at the third year. Therefore, SARS patients may become susceptible to the same virus 3 years after recovered from the infection[2], highlighting the importance of evaluating the durability of the humoral immune response to SARS-CoV-2.

The antibody responses against SARS-CoV-2 in humans are induced by some viral proteins, including spike glycoprotein (S protein) and nucleocapsid protein, among which S protein can induce neutralizing antibodies that are indispensable for viral neutralization and elimination, through blocking viral binding with host cells[1]. Similar to SARS-CoV-1, SARS-CoV-2 enters host cells via the binding with S protein to angiotensin-converting enzyme 2 (ACE2)[3], which is expressed on the surface of human alveolar epithelial cells, small intestinal epithelial cells, endothelial cells, and arterial smooth muscle cells[4]. SARS-CoV-2 S protein has an approximate size of 180 kDa, consisted of S1 and S2 subunits, the former of which contains ACE2 receptor-binding domain (RBD, amino acid residues 331–524)[5]. Anti-RBD IgG (RBD-IgG) titers have been shown to be strongly and positively correlated with virus neutralization[6]. Although highly homologous amino acid sequences are shared between the RBD regions of SARS-CoV-2 and SARS-CoV-1, the plasma of convalescent SARS patients or SARS-CoV-1 RBD monoclonal antibodies could not neutralize SARS-CoV-2, indicating the limited cross-neutralization protection between these two viruses[5,7]. Nevertheless, successful convalescent plasma therapy for COVID-19 patients has been reported: The symptoms of 10 severe COVID-19 patients who received 200 mL of convalescent plasma containing high-titer neutralizing antibody were significantly improved or even completely disappeared within 3–7 days[8].

Furthermore, it has been reported that most COVID-19 patients could produce virus-specific IgM, IgA, and IgG antibodies within a few days after infection[1]. According to a longitudinal study, though both IgM and IgA antibodies are produced early within 1 week after symptom onset, IgM reaches the peak at the 10th-12th days but the level subsequently decreases after 18 days, while IgA response persists at a higher level for a longer time period, reaching the peak at the 20th-22nd days[9]. On the contrary, the level of IgG antibody keeps increasing for 3 weeks after symptom onset, declines after 8 weeks, while remains detectable over 8 months[1,10,11]. However, the antibody response and neutralizing activity in COVID-19 convalescent patients up to 12 months are still unclear. For preventing and controlling SARS-CoV-2, as well as the vaccine development, the duration of functional neutralizing antibody response after individual infection with SARS-CoV-2 and the protective immunity for reinfection are necessary to be investigated by a long-term study.

Therefore, we aim to investigate the RBD-IgG response of convalescent COVID-19 patients for up to 12 months. In this study, a total of 1782 convalescent plasma samples from 869 COVID-19 convalescent plasma donors are tested for the presence and titers of RBD-IgG, which is proved to be positively associated with neutralizing antibody titers. In addition, influences of other factors (gender, age, and blood type) on the kinetics of RBD-IgG responses are analyzed. Our finding is critical for assessing the durability of the protective immunity induced by COVID-19 vaccines and predicting the future trend of COVID-19 pandemic.

## Results

**Sustained RBD-IgG responses to SARS-CoV-2 RBD.** In order to study the SARS-CoV-2 RBD-IgG responses in convalescent COVID-19 patients, 1,782 plasma samples obtained from 869 COVID-19 convalescent plasma donors in Wuhan are analyzed. All the samples are collected within 12 months after diagnosis. The presence and titers of RBD-IgG in the plasma samples are tested by using a CE-marked SARS-CoV-2 IgG enzyme-linked immunosorbent assay (ELISA) kit[12]. Figure 1A shows the positive rate of RBD-IgG in the convalescent plasma donors at different time points within 12 months after diagnosis. Specifically, a total of 390 plasma samples from plasma donors at the 1st and 2nd months after diagnosis (defined as the early stage following diagnosis) are tested, and the positive rate of SARS-CoV-2 RBD-IgG is 94.6% (the RBD-IgG titer is 1:80 or higher). At the 6th and 7th months after diagnosis (defined as the middle stage following diagnosis) and the 11th and 12th months after diagnosis (defined as the late stage following diagnosis), the positive rates of RBD-IgG are 89.4% and 81.2%, respectively. The titers of RBD-IgG are categorized as 1:80 (the actual value range is greater than or equal to 80 but less than 160), 1:160 (the actual value range is greater than or equal to 160 but less than 320), 1:320 (the actual value range is greater than or equal to 320 but less than 640), 1:640 (the actual value range is greater than or equal to 640 but less than 1280), 1:1280 (the actual value range is greater than or equal to 1280 but less than 2560) and 1:2560 (the actual value range is greater than or equal to 2560). Titers less than 1:80 are considered as negative, 1:80-1:160 as low titers, 1:320-1:640 as moderate titers, and 1:1280 and ≥ 1:2560 as high titers[13]. As shown in Fig. 1B, the proportion of higher titers displays a time-dependent downward trend, while the proportion of negative titers shows the opposite. Specifically, the proportions of plasma samples with moderate and high titers of RBD-IgG are 72.6%, 41.3%, and 27.2% at the early, middle, and late stages following diagnosis, respectively (Supplementary Table 1). It should be noted that even at the early stage following diagnosis, the titers of RBD-IgG in 5.4% of the convalescent plasma donors are at a level (<1:80) (Supplementary Table 1), which might be attributed to the failure of antibody production during infection or the low sensitivity of the method for antibody detection. As expected, at the late stage following diagnosis, the proportion of negative titers is increased to 18.8% (Supplementary Table 1).

**Longitudinal analysis of RBD-IgG responses.** We further determine the kinetics of RBD-IgG in COVID-19 plasma donors within 12 months after diagnosis. Monthly RBD-IgG titers are calculated and shown as geometric mean titers (GMT). As displayed in Fig. 2A, the RBD-IgG titers in convalescent plasma donors decrease within 12 months. After 9 months, the RBD-IgG titers begin to stabilize at a GMT of approximately 200. The RBD-IgG titer at the 12th month (a GMT of 189) following diagnosis is 69.9% lower than that at the 1st month (a GMT of 628) following diagnosis (Fig. 2A). As aforementioned, the antibody response

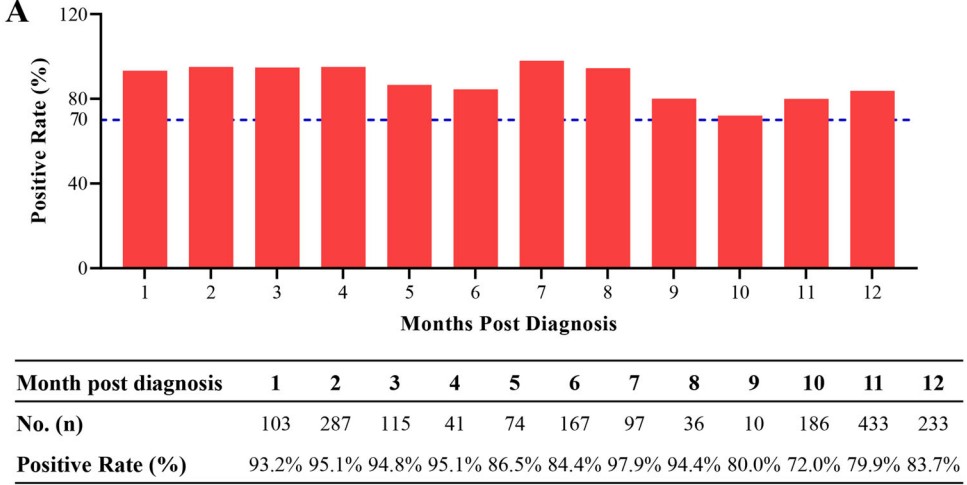

| Month post diagnosis | 1 | 2 | 3 | 4 | 5 | 6 | 7 | 8 | 9 | 10 | 11 | 12 |
|---|---|---|---|---|---|---|---|---|---|---|---|---|
| No. (n) | 103 | 287 | 115 | 41 | 74 | 167 | 97 | 36 | 10 | 186 | 433 | 233 |
| Positive Rate (%) | 93.2% | 95.1% | 94.8% | 95.1% | 86.5% | 84.4% | 97.9% | 94.4% | 80.0% | 72.0% | 79.9% | 83.7% |

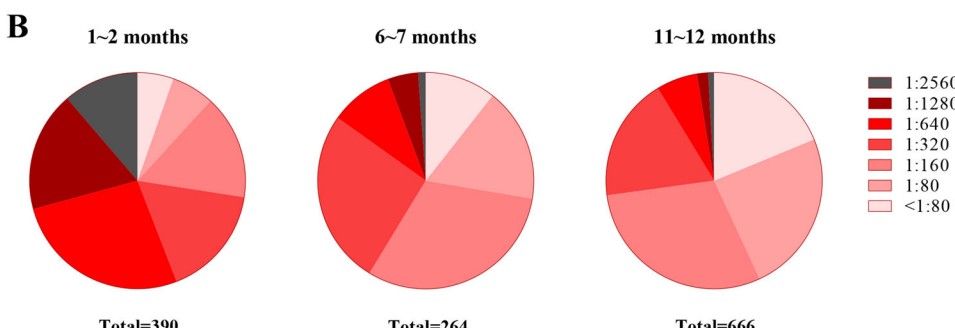

**Fig. 1 RBD-IgG titers against SARS-CoV-2 over times. A** Percentage changes of positive RBD-IgG. **B** Changes of RBD-IgG titers distribution. The titers of RBD-IgG are categorized as 1:80 (the actual value range is greater than or equal to 80 but less than 160), 1:160 (the actual value range is greater than or equal to 160 but less than 320), 1:320 (the actual value range is greater than or equal to 320 but less than 640), 1:640 (the actual value range is greater than or equal to 640 but less than 1280), 1:1280 (the actual value range is greater than or equal to 1280 but less than 2560) and 1:2560 (the actual value range is greater than or equal to 2560). Titers less than 1:80 are considered as negative.

dramatically augments at the initial stage after SARS-CoV-2 infection, followed by gradual declining[1,10,11]. Since the plasma samples are collected after the discharge of the plasma donors from the hospital (no less than 3 weeks after symptom onset), the period of antibody generation and expansion during the initial stage of onset might not be included.

In addition, we further analyze the kinetics of RBD-IgG in 14 COVID-19 convalescent plasma donors who repeatedly donate plasma for 3 or more than 3 times. Consistent with the general kinetics of RBD-IgG for COVID-19 convalescent plasma donors, the titers of individual plasma clearly display a downward trend within approximately 300 days (Fig. 2B).

In order to evaluate the stability of RBD-IgG titers in COVID-19 convalescent plasma donors after a long period of time, we re-collect the plasma from 237 donors, who present different titer levels of RBD-IgG at the early stage following diagnosis, during the 10th and 11th months after diagnosis. It shows that the plasma donors with higher initial titers remained faster RBD-IgG attenuation (Fig. 3A). Interestingly, after a long period of time, the RBD-IgG titers of the plasma donors with higher initial titers are still higher than those with lower initial titers. From the 1st-2nd month to the 10th-11th month, the RBD-IgG titer is decreased by 71.4% to a GMT of 207 in the 237 plasma donors (Fig. 3A). Additionally, we analyze the changes of RBD-IgG titers in plasma donors stratified by the initial titers from the early stage to the 10th and 11th months. As shown in Fig. 3B and Supplementary Table 2, 51.67% of the low-titer population turn

into negative after a long period of time, while 60.75% of the moderate-titer population and 54.29% of the high-titer population are decreased by one titer grade. Nevertheless, 11.67% of the low-titer population and 1.87% of the moderate-titer population are increased by one titer level after a long period of time (Fig. 3B; Supplementary Table 2).

**Influences of other factors on RBD-IgG responses**. In addition, we also examine the influences of gender, age, and blood types of the plasma donors on their RBD-IgG responses. The early, middle, and late stages following diagnosis are analyzed for the generation and durability of RBD-IgG (Fig. 4). The average titer of RBD-IgG in the plasma samples from male plasma donors is significantly higher than that from females within 12 months, especially at the early stage following diagnosis (p < 0.0001); however, the significant difference of RBD-IgG titers between male and female plasma donors is compromised at the middle and late stages following diagnosis. Furthermore, we find that the RBD-IgG titers are consistently and positively correlated with the age of the plasma donors, displaying an elevated RBD-IgG titer along with an increased patient age (Fig. 4B). The above-mentioned correlations between RBD-IgG titer and patient gender/age are consistent with the findings from several previous studies[1,14,15].

Zhao et al. reported that during the outbreak of the epidemic in Wuhan, 1,775 COVID-19 patients exhibited a distribution of

**A**

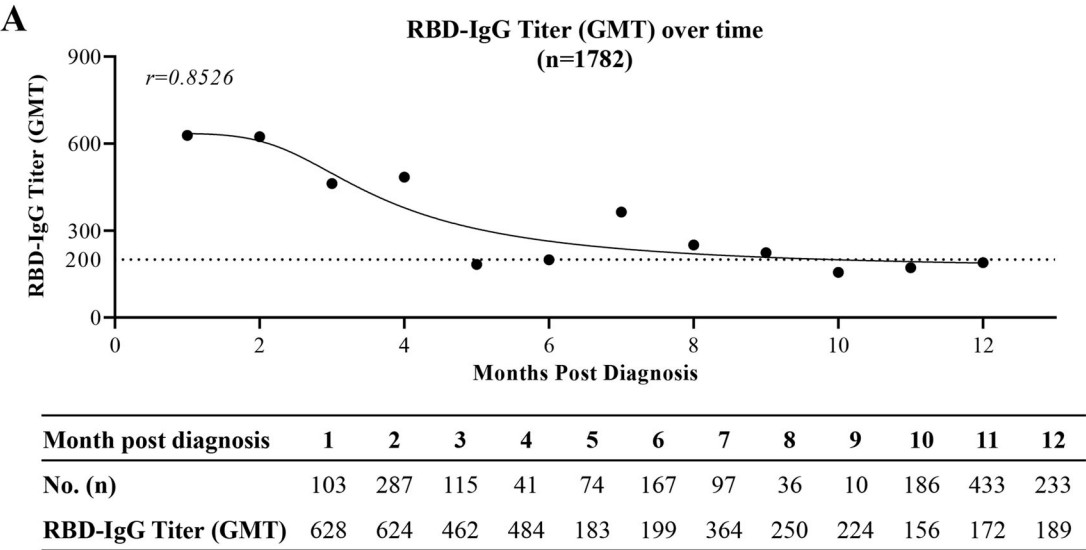

| Month post diagnosis | 1 | 2 | 3 | 4 | 5 | 6 | 7 | 8 | 9 | 10 | 11 | 12 |
|---|---|---|---|---|---|---|---|---|---|---|---|---|
| No. (n) | 103 | 287 | 115 | 41 | 74 | 167 | 97 | 36 | 10 | 186 | 433 | 233 |
| RBD-IgG Titer (GMT) | 628 | 624 | 462 | 484 | 183 | 199 | 364 | 250 | 224 | 156 | 172 | 189 |

**B**

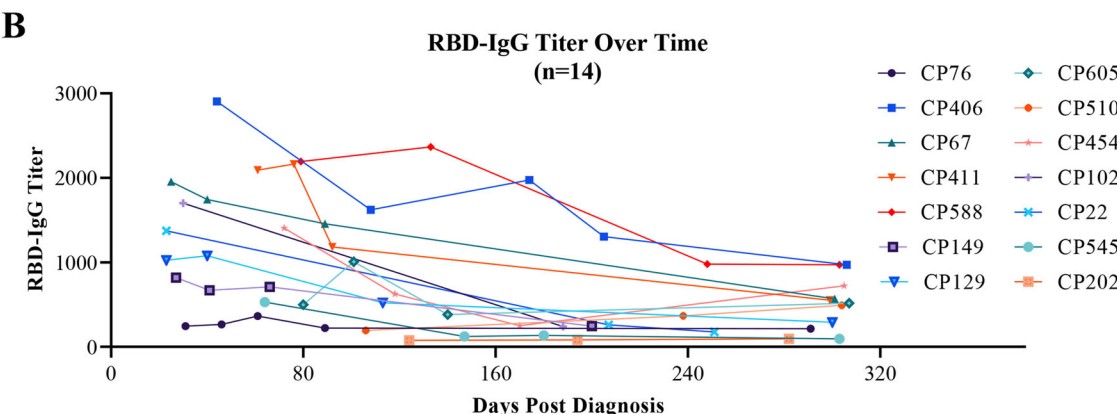

**Fig. 2 Kinetics of SARS-CoV-2 RBD-IgG responses. A** RBD-IgG kinetics curve of 1,782 plasma samples within 12 months. Monthly titer represents the GMT titer of all plasma samples collected in each individual month. GMT: Geometric mean titer. The value of RBD-IgG titer that was lower than the range (<1:10) was set to 1 for GMT calculation. **B** The RBD-IgG titers change in 14 COVID-19 convalescent plasma donors with multiple donations. CP76, CP406, CP67, CP411, CP588, CP149, CP129, CP605, CP510, CP454, CP102, CP22, CP545, CP202: the number of COVID-19 convalescent plasma donors. GMT, geometric mean titer.

37.75%, 26.42%, 25.80%, and 10.03% for A, B, O and AB blood types, respectively; the distribution in 3,694 normal people in Wuhan was 32.16%, 24.90%, 33.84%, and 9.10% for A, B, O and AB blood types, respectively[16]. These results indicate that individuals with A-blood type are associated with a higher risk of infection, while those with O-blood type are associated with a lower risk of infection and COVID-19 severity[16]. However, in our study, a negligible difference in RBD-IgG titers is found among the COVID-19 plasma donors of different blood types (Fig. 4C).

**Association between neutralizing antibody titers and RBD-IgG titers.** In order to confirm the correlation between RBD-IgG titers and the neutralizing activity of convalescent plasma samples, we conduct a plaque reduction neutralization test (PRNT) on Vero cells to define neutralizing antibody titers using an isolated viral strain[17]. The associations between the titers of PRNT (ID50) and RBD-IgG (dilution quantitative) based on a total of 150 COVID-19 convalescent plasma samples collected at the 1st-3rd months after diagnosis are analyzed. As shown in Fig. 5, the PRNT titers are positively correlated with the RBD-IgG titers ($r = 0.6657$, $p < 0.0001$).

## Discussion

The overall immunity to SARS-CoV-2, including the durability of immunity against the virus and vaccine-induced protective immunity, has not been fully understood. Neutralizing antibody response is crucial for the elimination of cytopathic viruses as well as preventing the reinfection of the virus[18]. The eliminating ability of neutralizing antibodies against SARS-CoV-2 in humans has been confirmed by clinical studies[8], whereas their protective effectiveness against the viral reinfection has only been proved based on animal models[19].

Effective vaccination is well known as the safest path towards herd immunity[1,20]. Most COVID-19 vaccine candidates employ the same antigen(s) of the original SARS-CoV-2, implying that the type and kinetics of neutralizing antibody response induced by vaccination are similar to those induced by the original live virus[21]. On the other hand, it has been reported that IgA and IgM antibodies are produced and reduced at the early stage of infection[9], raising concerns on the sustainable neutralizing activity of the patients' plasma. As a consequence, the long-term study of neutralizing antibody response in convalescent patients is essential to offer powerful support on developing COVID-19 vaccines and therapeutics.

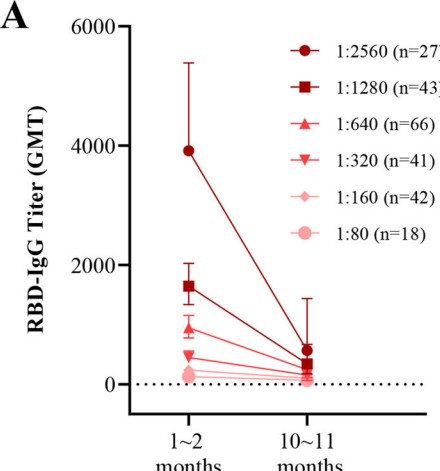

| Groups (Initial Titer) | No. (n) | 1~2 months (GMT) | 10~11 months (GMT) | Decreased (%) | Neg (%) |
|---|---|---|---|---|---|
| 1:2560 | 27 | 1:3916 | 1:567 | 85.5 | 3.7 |
| 1:1280 | 43 | 1:1646 | 1:345 | 79.0 | 0.0 |
| 1:640 | 66 | 1:947 | 1:247 | 73.9 | 7.6 |
| 1:320 | 41 | 1:451 | 1:154 | 65.9 | 22.0 |
| 1:160 | 42 | 1:237 | 1:103 | 56.5 | 45.2 |
| 1:80 | 18 | 1:129 | 1:69 | 47.3 | 66.7 |
| Total | 237 | 1:723 | 1:207 | 71.4 | 19.4 |

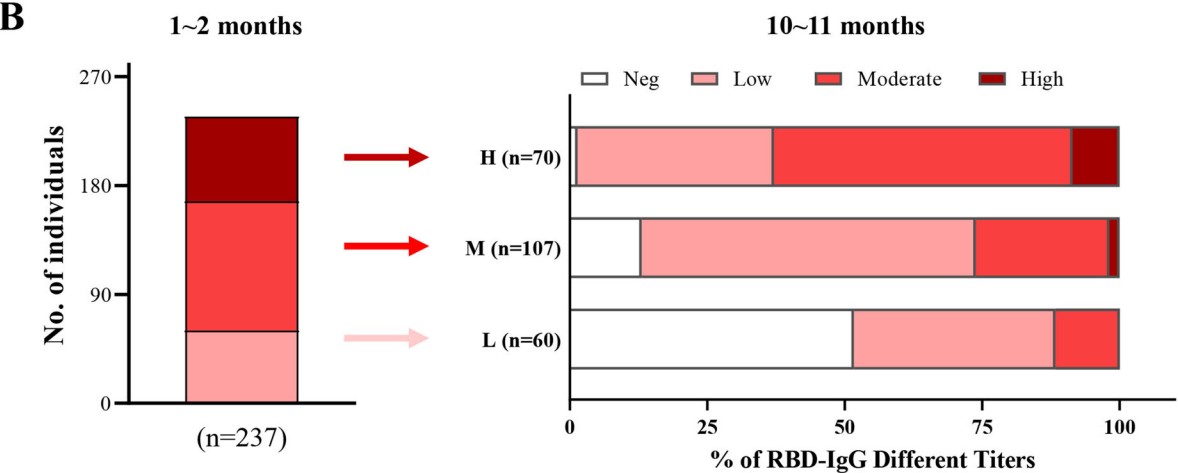

**Fig. 3 Stability of RBD-IgG titers over time. A** Changes of different RBD-IgG titers over time. The titer is expressed by Geometric Mean and Geometric SD. A total of 237 COVID-19 convalescent plasma donors with different titer levels of RBD-IgG at the early stage, who have donated plasma again at the 10th and 11th months, stratify with titers at the early stage following diagnosis, and the decline in titers and the proportion of plasma donors turning into negative are calculated. GMT: Geometric mean titer. Initial titer: RBD-IgG titer at the early stage (1st and 2nd months after diagnosis). **B** Change of RBD-IgG titers in plasma donors with low, moderate, or high titers after a long period of time. Neg (negative), titers less than 1:80. Low (low titers, L), titers of 1:80-1:160. Moderate (moderate titers, M), titers of 1:320-1:640. High (high titers, H), titers of 1:1280 and 1:2560. GMT, geometric mean titers.

At present, the longest observation period for SARS-CoV-2 antibody reaction kinetics is only several months[13,22–25], whereas no study has been carried out for more than 1 year time period. The titer of anti-SARS-COV-2 antibodies shows an overall decreasing trend over time, while several factors may accelerate the attenuation of anti-SARS-COV-2 antibodies. In a 6-month follow-up of COVID-19 patients in Paris, non-severe clinical presentation was found to be the only factor associated with the faster decay of IgG anti-spike antibodies[26]. However, the correlation between the titer change of neutralizing antibodies and clinical symptoms was not significant. Another 9-month study conducted in Wuhan showed that the titers of neutralizing antibody were not significantly decreased, regardless of symptoms[27]. Dispinseri et al. also reported that COVID-19 severity at disease onset of multiple co-morbidities cannot affect the kinetics and persistence of neutralizing antibodies[28].

To investigate the persistence of protective immunity against SARS-CoV-2, we have conducted a year-long kinetic analysis on SARS-CoV-2 RBD-IgG response in 1782 convalescent plasma samples obtained from 869 COVID-19 plasma donors, and assessed the constant influences of patient gender, age, and blood types on RBD-IgG response kinetics. Based on the fact of no further coronavirus disease outbreaks were found in Wuhan until the end point of plasma collection in this study, the 12-month immune response against SARS-CoV-2 in the COVID-19 plasma donors included in the present study is considered as primary immune response.

According to our findings, the RBD-IgG response in more than 70% COVID-19 convalescent plasma donors can persist for at least 12 months, indicating that vaccination can effectively restrict the spread of SARS-CoV-2. We will also examine the changes of RBD-IgG titers over a longer period of time. Compared with the titer obtained in the 1st month, the RBD-IgG titer is decreased by 69.9% in the 12th month. Moreover, the proportion of the plasma donors whose RBD-IgG titers remained at or above the moderate titer at the late stage following diagnosis is 27.2% (Supplementary Table 1).

Furthermore, we find that although the RBD-IgG titer gradually decreases over time within 12 months, the RBD-IgG titer is stabilized at a GMT of approximately 200 after 9 months

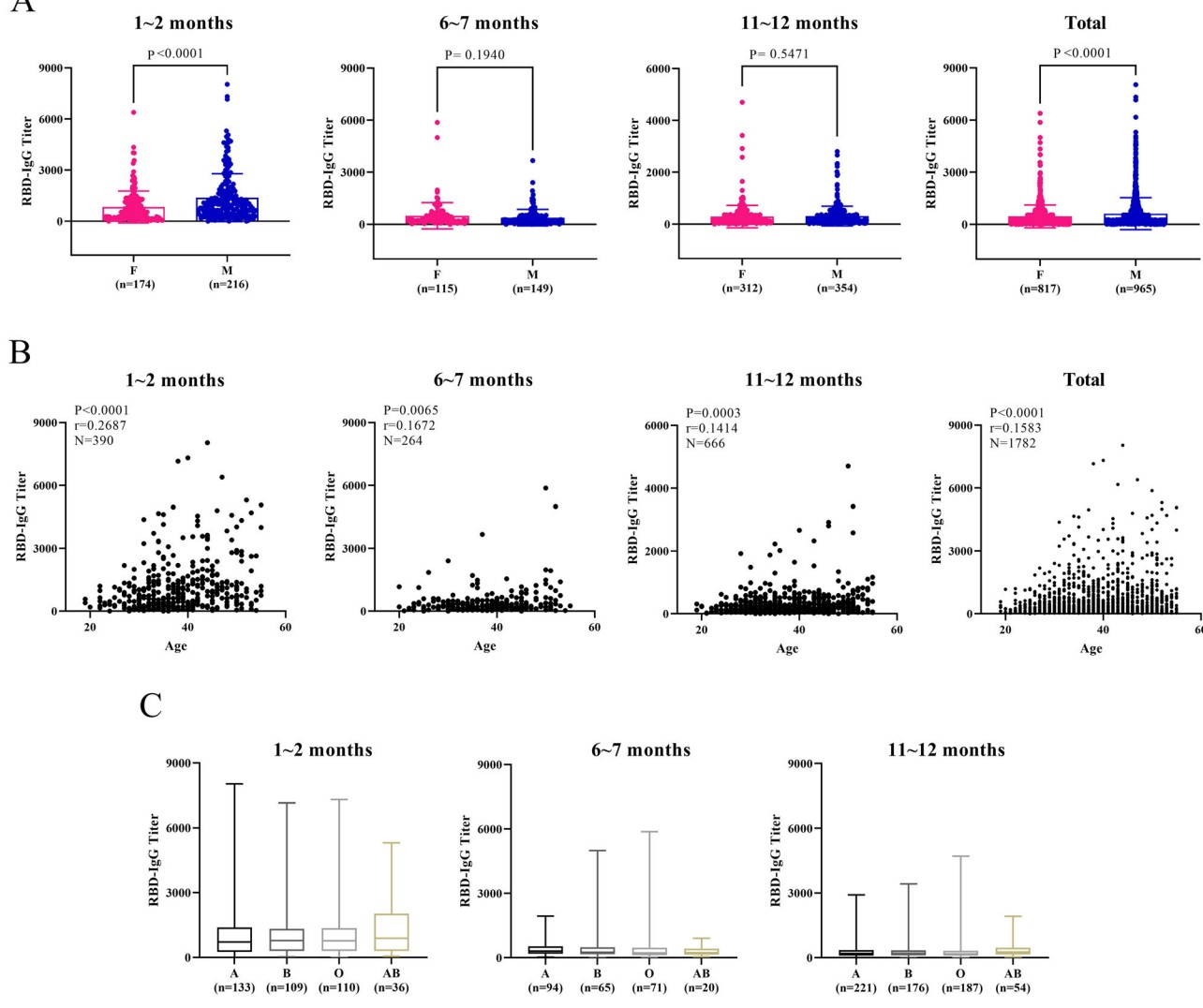

**Fig. 4 Influences of gender, age, and blood types on RBD-IgG response in COVID-19 plasma donors at different stages following diagnosis.** (**A**) is gender comparison (male compared with female). Two-way ANOVA and a two-tail unpaired t test with Welch's correction are used to test the difference between genders. Error bar in gender indicates mean with SD. F, female. M, male. (**B**) Age has a positive correlation with the RBD-IgG titer. Two-tailed Pearson correlation test was used to examine the association between age and RBD-IgG titers. (**C**) Insignificant difference was found between blood types of the plasma donors with the RBD-IgG titers. Box in blood groups indicates median and quartiles, while whiskers indicate maximum to minimum. A, B, O, and AB are Blood types.

following diagnosis. Considering that the half-life of IgG is around 21 days[29], the sustained persistence of RBD-IgG titer over time is probably produced by long-lived bone marrow plasma cells (BMPCs), which serve as the main source of protective antibodies[30]. In a 7-month study with 73 mild COVID-19 patients participation, S protein-specific long-lived BMPCs are detected in the COVID-19 convalescent patients at the 7th months following diagnosis[30].

We also evaluate the stability of RBD-IgG with different titers (stratification is based on the titer value at the early stage following diagnosis) after a long time period (the 10th and 11th months). Although more rapid attenuation of RBD-IgG is observed in the plasma donors with high titers, after a long time period, the RBD-IgG in plasma donors with higher initial titers remains to be higher than those with lower initial titers. Interestingly, we find that the RBD-IgG titers significantly increase in 11.67% of the low-titer population and 1.87% of the moderate-titer population during the 10th–11th months, which might be attributed to delayed seroconversion in a small number of plasma

donors. Similar phenomena have also been reported by Icahn School of Medicine at Mount Sinai[13,31].

Understanding the influences of factors in patient population, including gender, age, and blood type, on RBD-IgG response is critical to prevent SARS-CoV-2 infection, which can provide potential explanations for clinical symptoms. Therefore, we further correlate such factors of the COVID-19 plasma donors with their RBD-IgG titers, and find that the titer of RBD-IgG in male plasma donors is significantly higher than that in female plasma donors at the early stage following diagnosis. Consistent with previous reports, the positive correlation between age and RBD-IgG titer in a population of plasma donors aged 18 to 55 years indicates that elder plasma donors might develop antibody response against SARS-CoV-2 more effectively than younger ones[14]. Nevertheless, no significant correlation is identified between the blood type of the patients and their RBD-IgG titers, indicating that although individuals with A blood type are associated with higher risk for SARS-CoV-2 infection, and the susceptibility to SARS-CoV-2 is not related to RBD-IgG titers.

Last but not least, we perform PRNT on Vero cells to test the neutralizing antibody titers from 150 COVID-19 convalescent plasma samples, followed by analyzing the correlation between RBD-IgG and neutralizing antibody titers. Consistent with the previous report[6], our data show a strong positive correlation between these two types of titers (Fig. 5).

In conclusion, this 12-month longitudinal study demonstrates that despite of the downward trend of RBD-IgG response kinetics in COVID-19 convalescent plasma donors, over 70% plasma donors persist to produce RBD-IgG at detectable levels for longer than 1-year post diagnosis, which stably remains at a GMT of approximately 200. In addition, the RBD-IgG titers of male plasma donors are higher than those of female plasma donors at the initial stage of infection, meanwhile, age is positively correlated with the RBD-IgG titers. Furthermore, we confirm the positive association between RBD-IgG and neutralizing antibody titers. Overall, this study provides long-term strong support for the duration of protection by neutralizing antibodies in COVID-19 plasma donors, indicates the potential to prevent SARS-CoV-2 reinfection, and illustrates the role of neutralizing antibodies in clinical research and development evaluation of vaccines.

## Methods

**Ethics**. This study was approved by the Ethics Committee of Tiantan Biological R&D Center (Approve no. KY-EC2020-02), and each participant signed an informed statement.

**Donors for convalescent plasma transfusion**. From February 1, 2020 to January 10, 2021, 869 COVID-19 convalescent plasma donors in Wuhan, China were recruited, and 1,782 convalescent plasma samples were collected. Donors all met the criteria for release of isolation and discharge from the hospital according to the "Diagnosis and Treatment Protocol for Novel Coronavirus Pneumonia (Trial Version 4 and subsequent versions)", released by the National Health Commission & State Administration of Traditional Chinese Medicine. The donors were between 18 and 55 years old, with no history of blood-borne diseases, and screened by clinicians following blood donation standards. The interval between two plasma collections was no less than 14 days.

As shown in Fig. 6A, the average age of the donors was 38.21 years old (95% CI: 37.67-38.75). The proportion of plasma Rh-positive blood types (a total of 834 donors with blood type recorded) of A, B, O, and AB was 34.89%, 27.46%, 29.38%, and 8.27%, respectively (Fig. 6B).

**Plasma preparation and quality control**. Plasma preparation was conducted in accordance with the "*Protocol of Clinical Treatment with Convalescent Plasma for NCP Patients (2nd trial version)*", released by the National Health Commission & Bureau of the Logistic Support Department of the Central Military Commission (CMC). Plasma collection was carried out at least 3 weeks after the first symptom onset of patients.

**Detection of SARS-CoV-2 RBD-IgG**. CE-marked Wantai SARS-CoV-2 IgG kit (catalog number: WS-1396, Beijing WanTai Biological Pharmacy Enterprise Co., Ltd. Beijing, China) was used to test the titer of RBD-IgG. COVID-19 convalescent plasma. (assigned as 1:320, lot number: 2020021702, prepared by Sinopharm Wuhan Plasma-derived Biotherapies Co., Ltd.) was used as the reference standard. The target titer was obtained from 10 tests with Wantai SARS-CoV-2 IgG kit, which was expressed as the dilution and was defined by the largest dilution that optical density (OD) value is higher than the cut-off value of the kit. After validation of linearity, range, precision, the quantitative method was used to test samples. The reference standard 2-fold serially diluted with Wantai kit specimen dilution buffer at 6 concentrations (1:10, 1:20, 1:40, 1:80, 1:160, and 1:320). An aliquot of 100 μl diluted standards was added into Standard wells. An aliquot of 100 μl properly diluted samples was added into the wells of RBD antigen-coated plates, and incubated at 37 °C for 30 min. Each well was washed 5 times with diluted Wash Buffer. Subsequently, 100 μl of HRP-Conjugate was added to each well except the Blank well, followed by the same incubation and washing procedures. Then, 50 μl of Chromogen Solution A and 50 μl of Chromogen Solution B were added into each well, gently mixed, and incubated at 37 °C for 15 min in

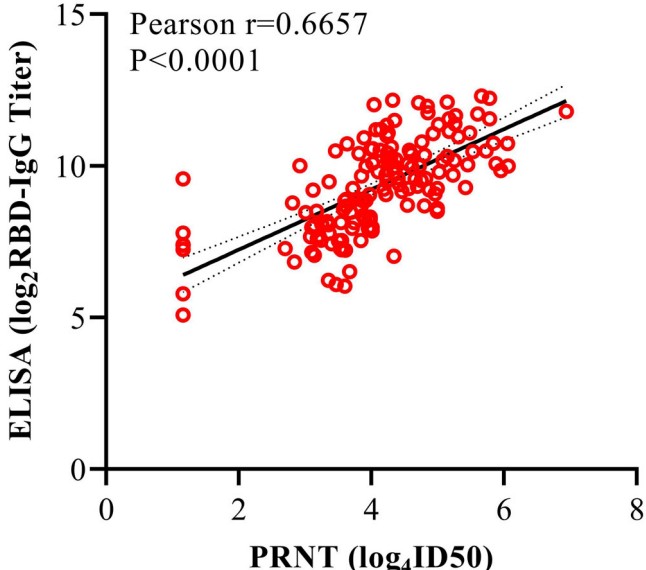

**Fig. 5 Neutralizing activity of plasma samples positively correlated with RBD-IgG titers.** Neutralizing antibody titers are transformed by Log$_4$ and RBD-IgG titers are transformed by Log$_2$. The association between neutralizing antibody titers and RBD-IgG titers is assessed by simple linear regression analysis and two-tailed Spearman's correlation. Spearman r = 0.6657 (95%CI, 0.5658 to 0.7463). It represents the 95% confidence bands with the best-fit line. PRNT, plaque reduction neutralization test.

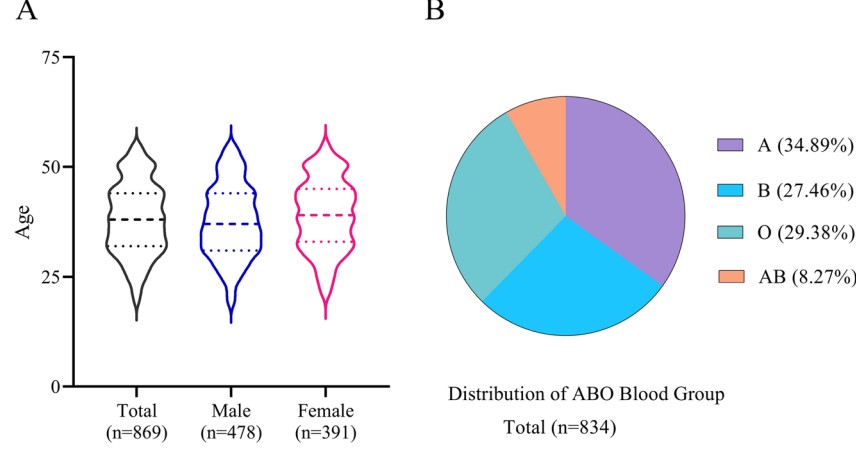

**Fig. 6 Distribution of age, gender, and ABO blood group in COVID-19 convalescent plasma donors recruited in this study.** (**A**) A total of 869 donors with 478 males and 371 females participated in this study. The average age of the donors was 38.21 years old (95% CI:37.67-38.75). The lines in the violin represent median and quartiles of age. (**B**) Distributions of 34.89%, 27.46%, 29.38% and 8.27% for A, B, O and AB blood groups. A, B, O, and AB are Blood groups.

darkness. Finally, 50 μl of Stop Solution was added to each well and gently mixed. The absorbance in each well was measured at 450 nm using SpectraMAX plus384 (Molecular Devices, Silicon Valley, CA, United States), and the standard curve was generated by SoftMax5.2 software. Use the dilution as abscissa, the OD value as ordinate, and the SoftMax5.2 to formulate the four parameters-equation standard curve. The acceptable criteria for linearity of the standard curve were: the $R^2$ value > 0.99; the recovery rate of each standard dilution ≤ 20% of the target value. According to the OD value of the sample, the corresponding titer is found in the standard curve. Samples, with results exceeding the upper limit of detection will be diluted again to obtain accurate results.

**Plaque reduction neutralization test (PRNT)**. Neutralizing titers were defined on Vero cells (ATCC CCL-81.4) using SARS-CoV-2 virus (WIV04 strain, IVCAS 6.7512) by PRNT. WIV04 strain isolation and its complete genome sequence were described previously[32]. All the plasma samples were inactivated at 56 °C for 30 min before using. The samples were initially 20-fold diluted, and then 4-fold serial dilutions were prepared in maintenance medium. The virus suspension (0.3 ml of 600 plaque-forming units [PFU]/ml) was mixed with an equal volume of the serum at desirable dilutions and incubated for 1 h. The mixture was added to Vero cell monolayer in 12-well plates and incubated for 1 h. Following removal of the mixture, 2 ml of the maintenance medium containing 0.9% methylcellulose (Sigma-Aldrich, Inc., St. Louis, MO, USA) was added to each well. The plates were incubated in a 5% $CO_2$–air incubator at 37 °C for 3–5 days. The neutralizing titer was calculated as the reciprocal of the highest antiserum dilution suppressing 50% of plaque forming.

**Statistical analyses**. GraphPad Prism 9.0 was used for statistical analysis. As for GMT calculation, the value of RBD-IgG titer that was lower than the range (<1:10) was set to 1. The influences of gender and blood type on RBD-IgG titer were analyzed by two-way ANOVA for inter-group difference, and by a two-tailed nonparametric Mann–Whitney $t$ test for median difference between groups. Unpaired $t$ test with Welch's correction was adopted for further analysis when a significant mean difference was found between genders. Two-tailed Pearson correlation test was used to examine the association between age and RBD-IgG titers. Neutralizing antibody titers were $Log_4$-transformed, RBD-IgG titers were $Log_2$-transformed, and linear regression analysis and Spearman's correlation were used to examine the association between neutralizing antibody titers and RBD-IgG titers. A P-value less than 0.05 was considered significant.

**Reporting summary**. Further information on research design is available in the Nature Research Reporting Summary linked to this article.

## Data availability
The authors declare that the data supporting the findings of this study are available within the paper and its supplementary and Source files. Other data that support the findings of this study are available from the corresponding author upon reasonable requests. Source data are provided with this paper.

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

## Acknowledgements
The authors thank all the plasma donors for participating in this study. This study was funded by the key project of "Preparation of specific plasma and specific globulin from patients with a recovery period of COVID-19 infection" (No. 2020YFC0841800) in the special project of "Public Safety, Risk Prevention and Emergency Technical Equipment" by the Ministry of Science and Technology China.

## Author contributions
X.M.Y., H.C.Y., D.X.F., Y.L.H., and D.B.Z. conceived and designed the study. C.S.L., D.Y., and H.L. contributed to implementation and management of the study. XXZ revised the study design and critically reviewed the manuscript. F.P.L., H.H.G., M.M., Y.G., R.Z., and H.R.N. collected the donor samples under the supervision of X.W., D.M.D., Y.X., and T.D. Z.J.Z. and L.Z.L. performed ELISA experiments under the supervision of Y.H. X.Z., Y.Z., and K.D. performed ABO Blood group tests under the supervision of J.H.Y. J.L. performed PRNT experiments under the supervision of Z.J.W. and K.D. L.F., F.F.W., S.L.Y., and Y.P. analyzed and interpreted the data. J.Z.W. and Y.Z. drafted the manuscript. T.J.L. generated and finalized the figures under the supervision of H.L. All authors reviewed the manuscript and provided critical feedback on the manuscript.

## Competing interests

The authors declare no competing interests.
