## [Peer Review File · Nature Communications]

REVIEWERS' COMMENTS

Reviewer #1 (Remarks to the Author):

Summary: A twelve months longitudinal study following 1 a total of 1,782 plasma samples from 869 convalescent plasma donors in Wuhan using a FDA approved CE-marked coronavirus RBD-IgG antibody detection kit from Beijing WanTai Biological Pharmacy Enterprise Co., Ltd. (Beijing, China. The test is only limited to the qualitative detection of anti-COVID-19 antibody in plasma and has a sensitivity and specificity of 96.7% and 97.5% respectively with no cross reactivity to HIV according to the manufacturer.

The authors report that donors, over 70% plasma donors persist to produce RBD-IgG at detectable levels for longer than 1 year post diagnosis, higher titers of male plasma donors compared to female plasma donors, a downward RBD-IgG trend over time, positive correlation of titers with age and neutralizing antibody titers. No correlation with blood groups were observed. Overall, this study contribute with urgent information about adaptive antibody responses to Covid19 and are largely in agreement with several related articles with shorter longitudinal studies (<6months).

Novelty: The authors assess the RBD-IgG antibody detection by using an approved ELISA kit (WanTai) with excellent performance. However, considering recent increase in mutation rate of spike proteins, potential problems could appear with such tests. If possible, perhaps authors could comment if such event could affect the test they are using?

This study does add crucial data to the field.

Appropriate statistics: Yes, fairly adequate sample size and statistics performed.

Referenced work: Is appropriate, however, as the filed is rapidly moving, please make sure most recent publications now available online are included in reference list (ex. Dynamics of SARS-CoV-2 neutralising antibody responses and duration of immunity: a longitudinal study - March 23, 2021 [https://doi.org/10.1016/S2666-5247\(21\)00025-2](https://doi.org/10.1016/S2666-5247(21)00025-2)).

Reviewer #2 (Remarks to the Author):

This study provides insight in antibody titers against Sars-Cov-2 in convalescent plasma donors over a 12 month period.

Pro:

large number of donors (>800)

long follow-up (12 months)

Con:

few individual profiles of Ab vs time

only anti-RBD

Several aspects are unclear. It should be better explained how different titers were determined/calculated. For the IgG elisa, it appears that a calibrator was used. Are the reported titers actually titers or units read from this calibrator? If titer, then the plateau that appears to be reached

after 9 months is ca. 1 dilution above the cut-off; does this impact on the plateauing that is observed? Or would this not be observed with a more sensitive assay? In Figure 5, the total variation in IgG titers seems to be roughly a factor of 2... Whereas the data shown in the previous Figures ranges over 30-fold. Is this correct? And if so, why was such a small range tested?

The study should be seen in light of other studies about the longitudinal trends in Sars-Cov-2 antibodies over time in convalescent plasma donors. In that respect, this study reflects a relevant if somewhat incremental contribution to existing knowledge.

I would suggest that the authors make a better effort in citing relevant literature on this subject and discuss their findings relative to those studies.

REVIEWERS' COMMENTS

Reviewer #1 (Remarks to the Author):

Novelty: The authors assess the RBD-IgG antibody detection by using an approved ELISA kit (WanTai) with excellent performance. However, considering recent increase in mutation rate of spike proteins, potential problems could appear with such tests. If possible, perhaps authors could comment if such event could affect the test they are using?

Thank you for your reminding. The genetic variation of SARS-CoV-2 has appeared in the COVID-19 pandemic. Whether the antibody test reagents developed in the early 2020 are still effective for monitoring the antibody of the patients infected with mutant SARS-CoV-2 strains is unclear. Since there is still no second outbreak in China after the epidemic has been controlled since early April 2020, we could not collect samples from the patients infected with mutant SARS-CoV-2 strains, and none of the patients participating in this study reported to have been infected with mutant SARS-CoV-2 strains. We will continue to pay attention to the detection efficacy for S protein mutation in future studies.

Referenced work: Is appropriate, however, as the filed is rapidly moving, please make sure most recent publications now available online are included in reference list (ex. Dynamics of SARS-CoV-2 neutralising antibody responses and duration of immunity: a longitudinal study - March 23, 2021 [https://doi.org/10.1016/S2666-5247\(21\)00025-2](https://doi.org/10.1016/S2666-5247(21)00025-2)).

Thank you for your comment. We have updated the list of references.

Reviewer #2 (Remarks to the Author):

Several aspects are unclear. It should be better explained how different titers were determined/calculated. For the IgG elisa, it appears that a calibrator was used. Are the reported titers actually titers or units read from this calibrator? If titer, then the plateau that appears to be reached after 9 months is ca. 1 dilution above the cut-off; does this impact on the plateauing that is observed? Or would this not be observed with a more sensitive assay?

COVID-19 convalescent plasma (assigned as 1:320, lot number: 2020021702, prepared by Sinopharm Wuhan Plasma-derived Biotherapies Co., Ltd.) was used as the reference standard. It was tested 10 times with WANTAI SARS-CoV-2 IgG kit to acquire the Target titer. The titer of the reference standard is expressed as dilution, and is defined by the largest dilution that OD value is higher than cut-off value of the kit. After validation of linearity, range, precision, the quantitative method is used to test samples. Use the dilution as abscissa, the OD value as ordinate, and the Soft Max5.2 to formulate the four parameters-equation standard curve. According to the OD value of the sample, the corresponding titer is found in the standard curve. Samples, with results exceeding the upper limit of detection will be diluted again to obtain accurate results.

To this study, WANTAI SARS-CoV-2 IgG kit is sensitive. Since the association between antibody titers and the threat to infection is still unclear, Our study aimed to monitor the antibody changes. We will also continue to pay attention to more sensitive detection reagents.

We thank the reviewer for the good suggestion.

In Figure 5, the total variation in IgG titers seems to be roughly a factor of 2... Whereas the data shown in the previous Figures ranges over 30-fold. Is this correct? And if so, why was such a small range tested?

Neutralizing antibody titers were transformed by Log_4 and RBD-IgG titers were transformed by Log_2 . So, it seemed to be roughly a factor of 2. The association between neutralizing antibody titers and ELIAS RBD-IgG titers was assessed by a total of 150 COVID-19 convalescent plasma samples collected at the 1st-3rd months after diagnosis, which is sufficient for performing the association assay.

The study should be seen in light of other studies about the longitudinal trends in Sars-Cov-2 antibodies over time in convalescent plasma donors. In that respect, this study reflects a relevant if somewhat incremental contribution to existing knowledge. I would suggest that the authors make a better effort in citing relevant literature on this subject and discuss their findings relative to those studies.

Thank you very much for this suggestion. The kinetics of anti-SARS-COV-2 antibodies in COVID-19 patients has always been focused, and we have added a paragraph to the discussion at lines 215-225, citing multiple recent relevant literatures.